# Re-Thinking Spatial Design in Homes to Include Means and Access Restriction with Material Impacts as Passive Suicide Prevention Methods: A Systematic Review of Design for Australian Homes

Michael Booth [1,*], Pushpitha Kalutara [2] and Neda Abbasi [3]

1   School of Engineering and Technology, Central Queensland University, Brisbane 4000, Australia
2   School of Engineering and Technology, Central Queensland University, North Rockhampton 4700, Australia; p.kalutara@cqu.edu.au
3   School of Engineering and Technology, Central Queensland University, Sydney 2000, Australia
*   Correspondence: m.booth@cqu.edu.au

**Abstract:** This systematic review analyses research that introduces commercial design applications that could be adopted for suicide prevention in homes. Furthermore, this literature review captures social, spatial and biophilic design methods to improve wellness in homes using environmental design psychology. Safety and human wellness frame this spatial design research that examines means and access restriction to improve home safety and prevent suicides. Suicide is a growing phenomenon that deserves specific attention to how environments can impact or restrict events. There is a substantial evidence base to evaluate suicide prevention methods used in high-risk environments of health and healing environments, workplaces and incarceration facilities. This review outlines design methods using spatial arrangement and material choices to improve human wellness in homes. The effects of biochemical reactions, such as those studied in toxicology, and stress are considered in this research to suggest material choices and applications in design to improve mental health in homes. Spatial designs for suicide prevention can guide various prevention measures, such as adopting means and access restriction and environmental design methods for wellness and considering impacts during lockdown periods. Environmental design psychology research supplies evidence for improved spatial arrangements in homes, with evidence showing that design applications can restore and improve mental health. This systematic review shows evidence for planning methods to prevent suicides considering both means and access restriction with considerable biochemical impacts from design. Design methods discovered by this systematic review will be considered for future studies and used within economic modelling to demonstrate design guidelines that improve wellbeing and support existing suicide prevention methods for Australian homes.

**Keywords:** environmental psychology; home design; spatial design; suicide prevention; value management

## 1. Introduction

This paper explores the question of suicide prevention via building design by reviewing existing research using a systematic review process. The issue of home suicides significantly impacts community function. This systematic review is conducted to discover building design methods in homes within these communities for suicide prevention. A significant gap in knowledge is shown to exist in this review considering home design psychology research for suicide prevention. Suicide prevention considering mental health for home design planning provides phenomenal benefits, with one in five Australians experiencing mental illness at some point following the COVID-19 pandemic [1]. Environmental experiences have potential to improve mental health [2] and quell negative

feelings that lead or contribute to ideation or intentional self-harm [3]. Environmental neuroscience research shows biochemical impacts from building materials [4] that cause physical and mental health impacts such as pain, stress or depression. The rationale for this home design research is to determine causes for biochemical reactions or suicide triggers resulting from home designs, such as confusion, stress, sickness, anxiety or depression. Design solutions promote better mental health in homes to support community interventions, health treatment and physical suicide prevention methods. Considering mental health and psychological impacts as adverse biochemical reactions can reduce mental illness, and mental health can be promoted in specially designed homes [5,6]. Strategies of injury and suicide prevention design are reviewed for commercial health and healing spaces [2,3,7], incarceration facilities, workspaces [8] and learning spaces [9].

Researching mental health benefits shows that environmental psychology designs improve productivity and wellbeing with improved health and recovery rates [2,10]. Mental health management using designs for suicide prevention is explored in recent health and healing design research [2,3,7,11] and demonstrates effectiveness for use in homes. Improving mental health in homes is important for designers to consider for our society. According to statistics in 2020-21 during the COVID-19 pandemic that considered 19.6 million Australians aged 16–85 years, over two in five, being 43.7% or 8.6 million people, had experienced a mental disorder at some time in their life [1]. Furthermore, it was reported that 21.4% or 4.2 million people had a mental disorder for at least twelve months, had experienced a mental disorder at some time in their life and had sufficient symptoms of that disorder in the twelve months prior to the survey [1].

Mental illness affects approximately 43.7% of Australians. In Australia, natural biophilic designs provide restorative effects from the positive cognitive interpretation of natural forms and shapes [12]. Complimentary stress reduction theory (SRT) and attention restoration theory (ART) show benefits for wellbeing in homes using biophilic, social and spatial designs [2,13–19]. Environmental design psychology considering SRT and ART shows positive impacts, and with it, designers can micro-manage adverse designs that cause biochemical stress impacts such as cortisol release [20]. Health impacts by 'routines of stress' can alter brain patterns with the continued release of chemicals such as cortisol, which are related to design impacts including odour, air quality, heat stress, mould, or allergies from material toxicology [6]. By considering environmental neurocognition and biochemical impacts [4,21], design applications such as anti-viral lighting in high density residential spaces can support mental health [22]. Value management can be useful for considering the cost/benefit variables [23] in the design stages of construction. For this research, we consider suicide prevention methods and demonstrate preliminary value estimates of methods based on research findings.

## 2. Research Methodology

The main strategy to solve the proposed research question is to broaden the existing literature. Narrative literature review is a good technique to explore the research topic, but a systematic review will create a strong base of methodological rigour leading to reliable findings [24]. Hence, the current study adopts a systematic review, which is an extension of a larger research project for suicide prevention in homes by incorporating social, spatial, biophilic and value management aspects to house design. Six objectives supply a systematic review framework for this suicide prevention method research, considering spatial design and material impacts. The objectives arise from a large research project that applied environmental psychology and physical impacts to re-design homes and improve psychological comfort to prevent, injury, self-harm or suicide events. Existing reviews show 'a gap in research knowledge for home suicide prevention' [25,26], and this systematic review compliments previous research findings [25,26] with a final cost/benefit design value modelling analysis. The objectives show the need for this systematic review and could have resounding societal benefits. They are listed in short below:

- Investigate the effect of intervention using building design for suicide prevention.
- Investigate the frequency and rate of mental health conditions in the population of Australia including models and statistics of suicide in homes.
- Establish supportive design guidelines for health improvements considering mental health impacts for home designs.
- Examine the impact theory of the physical and contributory causal factors related to the phenomena of suicide in homes.
- Determine suitable design solutions for addressing biochemical impact risks for a future cost/benefit economic analysis.
- Identify adverse design impacts for a future value management cost/benefit analysis as a supportive quantification analysis for suicide prevention guidelines.

This building design review finds evidence to advance methods to improve mental health [17,27–31] and prevent suicides. Environmental psychology, environmental neuroscience and biochemical environmental impact are reviewed for impact evidence with a Preferred Reporting Items for Systematic Reviews and Meta Analysis (PRISMA) systematic review diagram using Covidence review software, adopted [32] as shown as follows in Figure 1. While PRISMA follows a structured format enabling transparency and less bias, the method has been highly preferred in construction and engineering research [33,34].

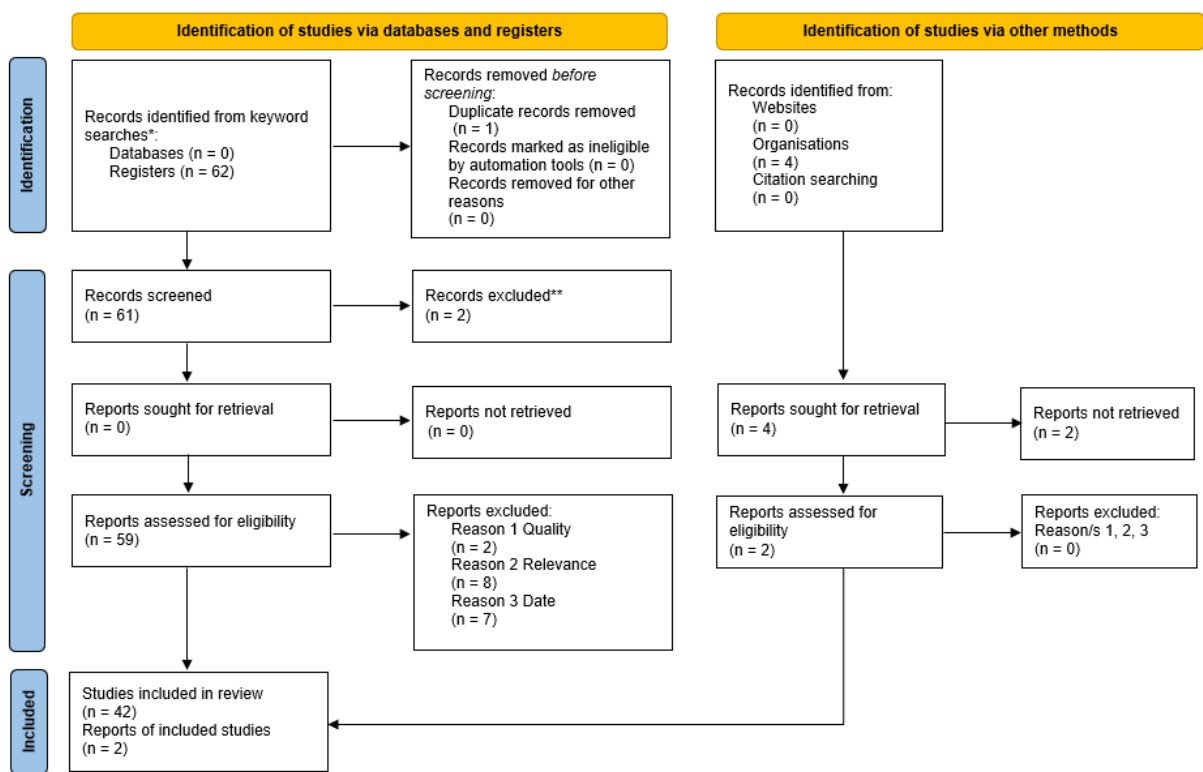

**Figure 1.** PRISMA systematic review screening diagram [32].

Exclusion search criteria** were sorted by relevance, date and topic significance for suicide prevention and mental health benefits. Inclusion of articles was conducted using the following search terms*: environmental psychology, suicide prevention, mental health design, means and access restriction, biochemical impacts, toxicology, biophilia, SRT and ART. This systematic review explored 64 articles gathered from academic databases including PubMed, ProQuest, EBSCOhost, ScienceDirect, Emerald, Wiley and PubMed for peer reviewed research in health and design methods. A total of 42 articles were included for this systematic review study, with 22 articles removed due to exclusion criteria related to quality, relevance, date and/or relevance. Research that offers key industry findings on the topic is sparse. Key findings include methods using designs addressing suicide and

that identify considerable design benefits to improve wellbeing and ameliorate feelings of hopelessness, suicide ideation and/or biochemical impacts from stress [20]. Future research will stem from topics identified from this research and combine theory findings to collate empirical-evidence-based design guidelines (EBDG).

## 3. Results and Discussion

This systematic review examines existing design evidence suitable for health improvement and suicide prevention including wellbeing design methods for homes. Mental health is a considerable part of environmental neuroscience and is investigated in the subfield of design psychology; neurotransmitter reactions to design can cause biochemical reactions such as stress, fear or confusion. Stress reactions cause cortisol release, whereas toxicological effects, e.g., pollutants or poor air quality, cause more physical impacts such as allergies and asthma [6]. Along with physical reactions to design, adverse environmental design psychology can cause or contribute to adverse mental health reactions such as fear, confusion, helplessness and anxiety.

Relative models of theory for suicide prevention by design were included in the review by [35]. These models include sociological theory (societal and community influences on suicidality); hopelessness theory (suicide as a fatalistic expectation of an individual); psychache theory (intense psychological pain and pain that can overpower any protective mechanism); escape theory suicide (failure and disappointment used to escape problems); and the interpersonal–psychological theory of suicide (feeling disconnected from society and burdensomeness). It also includes one relevant model considering how environments impact suicide: the stress diathesis model [35]. The stress diathesis model describes suicidal behaviour as influenced by individual biological/psychological predispositions, as well the surrounding environment [35]. Mental health design is included in health and healing settings for suicide prevention. The designs can be used in homes and are considerable for ameliorating feelings related to the stress diathesis model, hopelessness theory and interpersonal–psychological theory. Designs created to improve mental health in homes will supply recovery and restoration environments to boost recovery, as described by sociological theory, psychache theory, hopelessness theory, interpersonal–psychological theory and the stress diathesis model. Improving mental health designs in homes for suicide prevention by including biophilic, spatial and social aspects shows benefits across the literature [2,12]. Environmental design information from healthcare spaces [3] can reduce stress and ameliorate depression or suicidal ideation. Means and access restriction has also been reviewed [3,17,27,29,30] and is considered to be a useful physical strategy to prevent suicide events by considering mental health and biochemical and adverse physical reactions to built environments.

This systematic review research has been designed by collating physical restriction and physical and mental health design for risk areas in homes as a way to support existing prevention. Subsequent sections will provide a detailed description given to those critical aspects of home design and suicide prevention.

### 3.1. Spatial Design: Means and Access Restriction and Wayfinding

*"A persistent challenge for built environment design approaches to similar designs for means restriction applies to statistics that 75% of suicide deaths occur at home"* [29]

Barriers for suicide by jumping is an important suicide prevention strategy. Access to lethal means is included in suicide prevention literature regarding firearms, poisons and medications [35], and barriers for tall buildings including casual or video surveillance is also shown to be useful [17,31]. Means and access restriction complements mental health design and suicide prevention design in homes with tall spaces by offering solutions such as barriers and reduced capacities. Preventing suicide by hanging is shown to be more suitable for controlled environments and institutions, such as those of forced confinement, including psychiatric hospitals and prisons [35]. Safety is a structural objective of building design legislation such as the National Construction Code (NCC), and in this review, we

analyse the impact of designs after construction, considering safety, psychology and health impacts. Badly designed wayfinding causes confusion if complex and not user friendly, which can cause the loss of life in emergency events. Wayfinding is important for design to reduce mental health impacts. The mental health research conducted by Mackett (2021) shows that factors related to confusion, helplessness and anxiety are seen as threatening or uncomfortable experiences. Negative wayfinding design feelings can lead to future psychological sequalae for avoiding those systems [36]. Means and access restriction has been reviewed [3,17,27,29,30] and shown to be a useful physical strategy for preventing injuries and suicide events, with barriers, guards or fencing as effective methods of access restriction that can be supported by educational signage for support services in high risk areas [31]. Spatial design research, including spatial scales, shows benefits for spatial design to prevent injuries or jumping. Further health benefits stem from spatial design planning for safety and refuge, and privacy diagrams with biophilia show benefits for cognitive restoration [9,13] with spaces to escape, rest and restore mental cognition.

### 3.2. Biochemical Impacts: Physical and Mental Health

Biochemical impacts to human health and wellbeing can include adverse reactions to a built environment, often resulting from incompatible material choices, poor assembly or poor design methods. Air quality, sick building syndrome and odours are considerable causes for both physical and mental health impacts following prolonged exposure. Material impacts from adverse biochemical reactions in homes often arise in situations of water penetration and wet seal failure, which create chemical material decomposition factors.

A review of microbial aerosols states that the 'exposure to microbial aerosols is still common in many different environments and is often the cause of many adverse health effects' [37]. Toxic reactions from buildings are researched by Torgal (2012), who showed that the 'toxicity of buildings' has a variety of health-related material impacts for users. Biomaterial reactions result from wood preservatives; nanoparticles (insulation, cement and paint); and volatile organic compounds (VOC), including chemical carcinogens and endocrine disruptors [6]. Toxicity in buildings leads to health concerns from users over building material impacts, causing poor health from dangerous gases, particles or fibres emitted at room temperature. Materials such as carpet, linoleum, paint or plastic can decompose and become airborne, with older paint products containing lead and other materials containing radionuclides that can lead to ionizing radiation exposure [6]. Common VOC air pollutants that occur in indoor spaces include formaldehyde, benzene, xylene, acetaldehyde, naphthalene, limonene and hexanal. These pollutants can cause health effects such as eye and respiratory irritations, headaches and mental fatigue [6].

Environment impacts such as heat stress, climate and geographical design location can be considerable for design impacts on wellbeing. The results of a study conducted by Florido (2021) investigating heatwaves and relative humidity and their impacts on suicide (fatal intentional self-harm) showed humidity as more significantly related to suicide than heatwaves, with youth and women more significantly affected [38]. Several studies showed that 'there is a lack of benchmarking assessment criteria between the set-in put parameters and occupant behaviour for measuring the occupants' thermal comfort and assessing the overheating risk in a building' [39]. Findings show that designs for mitigating heat stress humidity can address patterns of poor mental health related to suicide. Daylight was shown to impact wellbeing [3], the circadian rhythm and melatonin (biochemical) release over time. The reviewed literature showed 'that windows and skylights confer benefits to home occupants through physiological and psychological mechanisms' [3]. Benefits are experienced by access to a view and increased daylight exposure [40,41], which can easily be included as base measures for design suitability.

Disease spread can cause stress and anxiety and impact mental health, and the reviewed literature considered the impact of COVID-19 on mental health with design methods that can mitigate and control disease spread by adopting improved materials. The impact of disease spread has caused significant detriment to mental health nationally and inter-

nationally, with lessons learned from the COVID-19 pandemic demonstrating low-cost design methods to mitigate and control disease spread by adopting improved systems and material applications. Ultraviolet lights and fittings can be used at entries and exits of public spaces and for high-density residential environments, such as lifts, foyers and exits, to decontaminate persons entering and leaving homes whilst controlling disease spread using design. Copper handles 'enable a reduction of the bacterial load on surfaces, in liquids and air' [22]. Automatic disinfection in publicly accessible surfaces, such as doorknobs and handrails, using material choices such as copper or brass doorknobs can help to reduce disease spread, improve safety and improve mental health. Copper/brass doorknobs can be installed to reduce disease and viral transmissions, with a review by Govind et al. (2021) showing the following:

1. Virus is active for 4 h on Copper surface.
2. Virus is active for 3 days on plastic/stainless steel.
3. Disease spread was minimized due to Copper/Brass door knobs.
4. Copper is preferred for doorknobs, push plates, handles, stair railings, restroom faucets and other applications of public places as Public surfaces are prone to disease-causing microbes.
5. Copper has antimicrobial properties [42].

The authors also stated that the 'Exposure of copper to COVID-19 is reported to inactivate viral genomes and showed irreversible impact on virus morphology, including envelope disintegration and surface spike dispersal' [42]. Designs using biomaterials to reduce disease spread in homes are promising methods to mitigate COVID-19 and thus improve health in homes. Lighting benefits for wellbeing can now be considered for design, including anti-bacterial lighting [22] in entrances and public spaces. Research by Rentfrow and Jokela (2016) [18] on geographical psychology showed that ecological influence contributes to geographical variation in psychological phenomena: 'The impact on gendered suicide per unit increase of heatwave counts ranges from −6.1 to +5% in suicide for males, and −6 to +6.8% in suicide for females' [18]. Considerable evidence indicates that features of natural and built environments, such as climate, terrain, green space and urban crowding, can affect individuals' psychological processes [18]. Further results showed that living near green spaces fosters wellbeing and reduces stress, and in geographical areas with high pathogen prevalence, individuals are more cautious and risk-averse behaviour is more common [18]. Architectural health design is covered in the systematic literature review conducted by Connellan (2013) for suicide prevention design planning methods considering stress. The review shows evidence of positive impacts for the following methods:

- Biophilic design
  - ➢ Gardens and art;
- Social design
  - ➢ Casual observation and connectedness;
- Spatial design aspects
  - ➢ Security, access restriction and natural lighting (circadian rhythms and chrono-biology).

Evidence on how interior design in healing environments improves mental health impacts focuses on user experience and includes post-occupancy evaluations [3]. Biochemical and psychological stress responses to environmental design, such as allergies, are natural stress responses produced to defend ourselves from harmful events or impacts. The complexity of complete home environment analyses provides limitations for this research; however, they are still relevant for inclusion in the economic modelling cost/benefit analysis of a larger future systematic literature review.

### 3.3. Environmental Design Psychology: Mental Health

Environmental psychology research demonstrates robust evidence for designs to improve mental health and prevent feelings associated with suicide. The literature shows the benefits of stress reduction, using design theory and stress reduction theory (SRT) to improve health and healing in environments [3] and to improve mental health in homes. Environmental design psychology for health spaces and aged care designs [43] can be used in home design to improve functionality. The environmental psychology theory of attention restoration theory (ART) can also provide benefits such as rejuvenation, healing, stress reduction [3] and increase cognitive function [12,44]. The literature shows benefits of improved mental health in homes during lockdown periods, when poor mental health correlates with increased injury events [30].

### 3.4. Value Management

Construction economics as value management (VM) planning of building projects and designs, provides the opportunity to improve benefits as presented in project life cycle and life cycle cost planning measurements. VM considers planning decisions for specified performance outcomes, such as legislative compliance and risk management. Value relates to design outcomes that are improved during planning and data analysis. Value can be considered for design changes, such as more detailed design drawings and changes in plants, assembly and construction methods, which will improve both cost and value outcomes such as efficiency, material durability and aesthetics [23]. VM provides the opportunity for issues analysis, risk management, functional design analysis, material compatibility, ethics, legal requirements and community considerations to suit design goals. Suicide prevention analysis can include VM cost/benefit measures for risk management in high-density residential projects [31]. VM planning can include preventative access measures and wellbeing considerations for community impact and risk management. Design risks for suicide and adverse mental health can be managed and prevented by including social, spatial and biophilic designs that can be evaluated via cost/benefit measures. Therefore, design guidelines were developed during the larger systematic review and evaluated by a VM analysis, with preliminary review findings listed in Table 1.

**Table 1.** Preliminary cost/benefit VM evaluation of design methods.

| Design Method | Cost (AUD) 1 to 5 | Suicide Prevention | Wellbeing Benefit (Mental Health) | Physical Prevention |
|---|---|---|---|---|
| Biophilia | 2 Low | Yes | Yes | No |
| Spatial design | 3 Medium | Yes | Yes | Yes |
| Means and access restriction | 2 Low | Yes | Yes | Yes |
| Social design | 1 Low | Yes | Yes | No |
| Environmental psychology | 2 Low | Yes | Yes | No |
| Legislation Suicide prevention evidence-based design guidelines (EBDG) | 4 High | Yes | Yes | No |
| Material–Biochemical impacts | 2 Low | Yes | Yes | No |
| Suicide prevention evidence-based design guidelines | 1 Low | Yes | Yes | No |

## 4. Systematic Review Findings

The reviewed literature shows evidence that complements existing suicide prevention methods with findings summarised in the subsequent paragraphs. Research findings across public spaces, including health, healing and building control environments, provide numerous physical suicide prevention methods that were discovered by previous home

design research [25]. Physical design methods are further considered in this systematic review resulting from research findings that show that spatial design can include means and access restriction in the value management (VM) planning of houses. Means and access restriction strategies are suitable for suicide prevention in high-density residential and urban planning settings, such as roof tops and car parks. Further consideration of design controls should be given for the physiological and biochemical impacts of home design on the path to zero home suicides. Suicide prevention that considers lethal access to jumping sites is useful for design planning and can supply low-cost planning solutions. Wayfinding should be considered in spatial designs where solutions bolster the environmental impact by improving mental health in dense housing spaces; this may provide another low-cost VM planning solution. This research discovered further evidence that VM planning designs for suicide prevention can also include the removal of biomaterials such as irritants, allergies, odours and contagions, analysed via life cycle cost analysis and toxicology, to improve both physical and mental health. Biomaterial design choices for suicide prevention methods provide future research benefits for considering health impacts and developing evidence-based design methods for cost/benefit economic modelling. Suicide prevention methods with cost/benefit values can consider both physical and mental health designs to improve homes in an effort to combat depression, stress and anxiety, which result from general home life and environmental impacts. By considering both material design choices and lethal means and access for suicide prevention in planning, we can improve life cycle costs, the quality of designs and the quality of life for users. Improving mental health designs for homes to improve psychological wellbeing using environmental design psychology is the future of home design and spatial analysis. With the health design solutions presented in this systematic review, which consider the complex variables of home designs to combat the issue of home suicides, it has been demonstrated that these planning aspects can benefit 44% of society.

## 5. Conclusions

This systematic review displays the significant knowledge gap in suicide prevention using building design, for which there are no design guidelines. Further design methodologies can be used to complement existing evidence-based environmental design guidelines for suicide prevention in homes. This review shows supportive evidence for the use of spatial, social and biophilic designs to improve mental health in built environments, with preliminary cost/benefit considerations for future study expansion and research development. This review displayed further information gaps regarding mental health design for homes, considering disease control and toxicology along with environmental psychology and biochemical impact variables. This review shows benefits for future cost/benefit home design economic modelling and VM planning. This suicide prevention research can help the greater community, improving low-cost design planning suicide prevention methods for homes.

**Author Contributions:** Conceptualization, M.B. and P.K.; methodology, M.B. and P.K.; software, M.B. and P.K.; formal analysis, M.B.; investigation, M.B.; resources, M.B.; data curation, M.B.; writing—M.B.; writing—review and editing, M.B. and P.K.; supervision, P.K. and N.A. All authors have read and agreed to the published version of the manuscript.

**Funding:** This research received no external funding.

**Data Availability Statement:** Any raw data used for this review can be provided upon reasonable request.

**Conflicts of Interest:** The authors declare no conflict of interest.

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
