# Peer review of "Re-Thinking Spatial Design in Homes to Include Means and Access Restriction with Material Impacts as Passive Suicide Prevention Methods: A Systematic Review of Design for Australian Homes"

_buildings, doi:10.3390/buildings13061452_

Round 1
Reviewer 1 Report
The paper presents interesting descriptive information on the biophilic design methods by reviewing the current state-of-the-art contemporary examples but it requires more justification on the selection of methodological approach for the study. The paper was set to execute the systematic literature review findings but there is no proper way of analysing the qualitative methods. I recommend to the authors to use the vosviewer visualisation tool to improve the credibility of the paper.
Please read this reference book - Ozarisoy, B. and Altan, H. (2023). Handbook of Retrofitting High Density Residential Buildings: Policy design and implications on domestic energy use in the South-eastern Mediterranean climate of Cyprus, Springer, https://link.springer.com/book/9783031118531
In order to improve the credibility of your work, I recommend you to cite this work properly into the manuscript.
Minor English editing is required.
Author Response
Thankyou for taking the time to review my work, Please find attached file in response.

Reviewer 2 Report
PAG5 paragraph 3.2 I don't understand the link between the main topic of the paper with the Covid-19
In general, the paper is a review that can open to further higher-quality research, that can be put in a special issue.
Author Response
Thankyou for taking the time to review my work, please find attached file in response.

Reviewer 3 Report
buildings-2394136 - Review v1
1. In this study, environmental psychology was applied to improve family health using spatial design method, which served as reference for suicide prevention. The results obtained during the pandemic were analyzed, which indeed has made significant contribution by providing useful information to relevant units.
2. In “2. Research Methodology”, six sub-questions were described. Further explanations regarding the sources of these six sub-questions should be provided.
3. Please provide further explanation on how the screening process in “Figure 1: Prisma systematic review screening diagram” was conducted. (e.g., how to search 61 articles through academic databases?) As the first part of this paper is focused on literature review, the screening process should be explained in more detail.
4. Is it possible to categorize the literature on spatial design based on the type of space, such as public spaces or hospital spaces in “3.1 Spatial Design”? or is it found that the literature only focused on specific spaces? For example, it is not possible to explore spatial design in private spaces.
5. In spatial design, "spatial scale" is a very important and crucial. Has the literature identified any specific spatial scales? If so, please include them. Otherwise, it should be listed as an important result and provided as a reference for future study.
6. In Section 3.2, it is apparent that clear answers can be obtained through measurement using various instruments. It is also an important issue for the physical environment of buildings and indoor health environments. If the relevant studies provide specific "numerical recommendations," they should be included in the paper.
7. In Section 3.3, are there any specific recommendations for environmental design in the literature? If there are, please include them.
8. It is recommended to present the results in a list form, including (1) recommendations for spatial design (qualitative descriptions), and (2) clear spatial numerical recommendations (quantitative values) if available. If there are no numerical recommendations available, they should be included as the recommendations for future study.
Author Response

(The authors gave the same response as above.)

Round 2
Reviewer 1 Report
The authors addressed all chages very thoroughly. Very well done.
I did not detect any issues on the grammar.